# The Use of Social Media and Its Impact on Shopping Behavior of Slovak and Italian Consumers during COVID-19 Pandemic

**Viktória Ali Taha [1,\*], Tonino Pencarelli [2], Veronika Škerháková [1], Richard Fedorko [1] and Martina Košíková [1]**

[1] Faculty of Management, University of Prešov, 08001 Prešov, Slovakia; veronika.skerhakova@unipo.sk (V.Š.); richard.fedorko@unipo.sk (R.F.); matina.kosikova@unipo.sk (M.K.)

[2] Faculty of Economics, University Urbino Carlo Bo, 61029 Urbino, Italy; tonino.pencarelli@uniurb.it

[\*] Correspondence: viktoria.ali-taha@unipo.sk; Tel.: +421-905-162-111

**Abstract:** The coronavirus crisis hit the world and affected all aspects of our lives, including consumers' habits, preferences, and shopping behaviors. The survey, which involved 937 respondents from two countries, examined how the pandemic affected shopping behavior and consumer preferences in Italy and Slovakia. This paper aims to explore the impact of social media on consumer behavior, more specifically, it examines the influence of social media on the preference of specific e-shops during the first wave of the COVID-19 pandemic. Spearman's rank correlation coefficient was used to determine a statistically significant relationship between the variables and the Mann–Whitney $U$ test and the Kruskal–Wallis $H$ test to assess the significance of differences between respondents in terms of demographic characteristics (residence, age, and gender). The results revealed the existence of statistically significant differences in the use of social media during the first wave of the COVID-19 pandemic in terms of various demographic factors as well as a relatively weak relationship between the social media used and the purchase in the e-shop promoted on the social media.

**Keywords:** consumer behavior; social media; COVID-19 pandemic and online shopping

## 1. Introduction

Due to a pandemic, 2020 was an extraordinary year. According to the World Health Organization, the COVID-19 pandemic "presents an unprecedented challenge to public health, food systems and the world of work". The pandemic brought destructive economic and social disruptions: extreme poverty and the loss of work and livelihoods to millions of people, existential problems for many companies and enterprises, border closures, trade restrictions, confinement, etc. [1]. Bradbury-Jones and Isham (2020, in [2]) argue that COVID-19 brought many significant psychological, social, and professional changes, such as physical and mental health problems, jobs lost, low savings, fear and stress during outside visits, and an uncertain future.

People and their lives have been strongly affected by the COVID-19 crisis in a number of ways. Various restrictions made by the governments and authorities affected peoples' health, and social and economic situation, as well as their attitudes and behavior. Due to movement restrictions and lockdowns, people remained isolated in their homes with limited contacts with others. Social media allowed them to connect with relatives, friends, schoolfellows, teachers, and/or colleagues. They also enabled people to have fun, distract themselves, and spend free time. In addition to that, social media platforms have played a crucial role in the dissemination of information and have been a rich and valuable source of information on COVID-19. These are probably the main reasons for the increased interest in using social media. For example, Italian social media users aged between 35 and 44 years spent 118 percent more time online during the first lockdown, followed by 45–54-year-olds with an increase in time spent online of 114 percent. In the 25–34 age group, the time spent online increased by 110 percent during the second week of complete lockdown [3]. According to Contenuti Digitali statistics about social media usage in Italy, there are 37

million active users of social media and networks in 2020. In 2020, four million new users were registered on social media in Italy where the biggest increase of them was observed on TikTok (475.1%), Pinterest (30.5%), Twitter (24.2%) and LinkedIn (19.5%). As many as 73 percent of Italians between 16 and 30 years stated they used social networks and instant messaging more. Based on observance of social distancing due to the COVID-19 situation, there was also increased daily use of messaging apps like WhatsApp (81%) and Facebook Messenger (57%) [4]. In November 2020, there were 3010 million active Facebook users in Slovakia (which, with a share of 84% of all social media visits, is considered to be the leading social media website in Slovakia). At the same time (November 2020), there were 1287 million Instagram users in Slovakia which represents an increase of 20% compared to January 2020 [5].

The pandemic has also been affecting consumer behavior. According to a McKinsey and Company study "the period of contagion, self-isolation, and economic uncertainty have changed the way consumers behave" and these new consumer behaviors cover all areas of life—the way people work, learn, communicate, travel, shop and consume, live at home, entertain themselves, and/or deal with health and wellbeing [6]. According to a PwC (a global network of firms delivering assurance, tax and consulting services) survey [7], COVID-19 rapidly reshaped consumer behavior, e.g., in the sense that consumers are buying more essentials (non-perishable groceries, household and cleaning supplies, frozen food, etc.) and taking advantage of shopping online.

During the pandemic, interest in online shopping increased significantly. The contactless shopping process made e-commerce the first choice for people when shopping; in connection with the boom in online shopping, many merchants decided to launch promotional activities through social media platforms to promote brands and increase sales through encouraging consumers to forward information and invite online friends [8]. Sabanoglu [9] sates that in Italy, the year-on-year increase in e-commerce turnover for the first quarter of 2020 was 20%. "Between February and March 2020, the online sales in Italy grew significantly compared to the same period in 2019. Particularly, during the weekend, the e-commerce sector was largely impacted by the outbreak of coronavirus (COVID-19). On March 8, online sales registered an increase by 90 percent compared to the same period of the previous year" [9]. In Slovakia, there was also a marked e-commerce turnover increase. Online sales between March 2019 and 2020 increased by 44%. E-shops with medical supplies in particular recorded a year-on-year increase in orders by 130% during the first lockdown due to increased interest in disinfectants and facial masks [10]. However, the situation was far from idyllic; while in some areas e-commerce flourished (e.g., groceries or hygiene items), others reported a significant decline in sales. According to a survey conducted by Casaleggio Associati, among Italian e-commerce companies in March 2020, more than 50 percent of the surveyed companies declared that they were negatively impacted by the pandemic [11]. The same phenomenon could be observed in Slovakia; a survey conducted on e-commerce companies revealed decreasing sales in 37% of all companies. The main problem faced by the Slovak e-shops was logistics problems and supply outages, as declared by representatives of Slovak e-shops. Among the sectors most affected by the coronavirus crisis is the tourism sector [12]. The largest Italian travel portal selling train tickets—Trenitalia, recorded a 65% decrease in sales in October 2020 compared to 2019. Their sales loss in 2020 due to the coronavirus crisis represents 2 billion euros [13]. Slovak online ticket and tour retailer Pelikan.sk recorded a drop in their website visit of approximately 580,000 visitors between January and March 2020, which caused a 70% decrease of potential clients [12].

Online shopping during the pandemic has its pitfalls, such as impulsive purchases. Naeem [2,14] presented interesting studies on how social media can shape fear and consumer responses during the COVID-19 pandemic, focusing on analysis of the role of social media to create panic behavior and thus a panic buying reaction. One of the main findings of his study is that "social proof and influence from close connections can enhance consumer panic buying behavior" (p. 7) thus confirming that social media forms people's

collective response to the coronavirus and shapes panic buying reaction [14]. The author points out rumors and disinformation on social media that have created perceptions of risk which subsequently led to customers impulsive buying in order to be able to stay at home for a longer period. He notes that the wave of social distancing and staying at home, as well as social media interactions and authorities' communications enhanced fear and uncertainty and led to impulsive buying [2]. Islam et al. [15] elucidate that "panic buying has become a global phenomenon reflecting that loss of control among consumers in the era of Coronavirus lockdown". Consumers' impulsive and obsessive purchases emptied store shelves, where there was a shortage of products, such as toiletries and groceries in particular. Information about peoples' panic purchasing and photos of empty shelves and long lines in front of stores have flooded social media platforms which contributed to more panic because those messages strengthen the consumers' tendency to buy impulsively and obsessively [15]. The above demonstrates the immense power of social media and its impact on consumer behavior, which only underlines the need to investigate this phenomenon, which was also the aim of this study.

## 2. Literature Review

Social media is rapidly and fundamentally changing; it is different from even a year ago [16]. It is defined as "internet-based channels allowing users to conveniently and selectively interact with each other and derive value from user-generated content" (Carr and Hayes, 2015, in [17]). Štefko and Steffek [18] stated that the past years have been a period of rapid growth in the virtual world; what is current and relevant today will be obsolete and out-of date tomorrow. The optimal use of new technologies, the Internet of things, virtual reality, artificial intelligence, and free stores that shape the picture of today will be a burdensome test for many retailers. Since there is no doubt about the power of social media and its importance for marketing activities, it has become a substantial part of marketing strategies of the companies [19].

Social media has had a significant role during the coronavirus crisis. Its influence and impact are visible in various areas of our lives—from work, through education and entertainment, to shopping. The COVID-19 pandemic and the associated social distancing and lockdown made individuals increasingly turn to social media for support, entertainment, and connection to other people. Social distancing and lockdown also changed how and what individuals self-disclose on social media [17]. Social media has enabled people to seek and share information about the pandemic and to maintain relationships with relatives, friends, and fellows during social distancing [15]. Nabity-Grover et al. [17] assume that the pandemic made individuals more aware of how they present themselves and what they disclose on social media during the pandemic—particularly as it pertains to their personal health, impact of their behavior on others, and perception of their views about health by others.

*The Use of Social Media during the COVID-19 Pandemic*

According advertising platform Criteo [20], which examined the behavior of Italian consumers during the lockdown, 61% of Italian consumers downloaded the online store application during the lockdown (38% of them used it on a daily basis), 59% of consumers downloaded the online food delivery application (26% of them used it on a daily basis and 26% several times a week), 46% of consumers downloaded the online food delivery application during the lockdown (24% of them used it on a daily basis and 29% several times a week), and 53% of Italian consumers stated that their social network use rate increased significantly during the lockdown, with 70% of them stating that they used social media several times a day. Hence, it would be relevant to investigate if there are some differences in the use of social media between Italian and Slovak consumers. In view of the above, the first hypothesis was formulated:

**Hypothesis 1 (H1).** *There are statistically significant differences between Italian and Slovak consumers in the use of social media (in terms of the type and the intensity of its use) during the first wave of the COVID-19 pandemic (lockdown).*

*Gender Differences in the Use of Social Media during the COVID-19 Pandemic*

An important characteristic that determines consumer behavior (not only in times of pandemic) is gender. Several authors in their studies pinpoint the gender differences and diverse intentions in using social media. According to the study by Krasnova et al. [21], the use of social media by women is driven by keeping social ties with their close and distant acquaintances, access to social information and positive attitude towards information sharing compared to men. Men's use of social media is motivated, on the other hand, by gaining information of a general nature, because they do not perceive commitments to be as important as women do. In general, while men perceive social media as a tool for expanding their weak social ties, women tend to strengthen their bonding relationships [22]. From another point of view, women show stronger perception in ease of use, relative advantage, and compatibility in using social media compared to men [23]. When we shed a light on perceiving privacy risk on identity information sharing on social media (e.g., full name, date of birth, home address, bank account number, phone number, etc.), women show significantly higher fears of their misuse which could lead to their unwillingness to realize an online purchase. Considering privacy risks in image posting, it was found that women perceive higher risk comparative to men when posting images especially on Snapchat and Reddit, while Facebook and Twitter they considered as more trustworthy media [24]. This phenomenon could be associated with the purpose of social media using in the point of relations view by each gender. When we take an insight on gender differences among Italian users, in the individual social media preference, it was revealed that the use of Snapchat (72% of all users) and Pinterest (76% of all users) is absolutely dominated by women. The main purpose of Snapchat lays in communication by images and relationship building, while Pinterest focuses on inspirations linked to propensity to online shopping [4]. Another study proved that photo and video content on social media influence more frequently women's decisions than men's [25]. When it comes to Generation Z, adolescent girls spend their time on social media and text messaging through their smartphones more frequently comparative to boys, who are more focused on gaming and electronic devices [26]. Especially in social media usage, they have the highest expectations for integrity (which is reflected in an increased level of trust in social media) and strong identification with published content [27]. The study by Hou et al. [28] revealed the increased prevalence of depression and anxiety among the Chinese population during the COVID-19 pandemic, while females are experiencing more severe anxiety symptoms than males. The authors also researched social media as the source of information related to COVID-19 and found that social media was the main source of updates on the COVID-19 related information. Although there were no gender differences in time spent on seeking information about COVID-19, there were differences in the rate of using traditional media as main source of information. The time spent searching for information on COVID-19 also proved to be an important factor, with respondents who spent more time searching for information (more than 60 min) having more severe symptoms of anxiety than those who spent less time. Based on the above, we assumed the existence of statistically significant differences in the use of social media between men and women and formulated the following hypothesis:

**Hypothesis 2 (H2).** *There are statistically significant gender differences in the use of social media during the first wave of the COVID-19 pandemic (lockdown).*

*Generational Differences in the Use of Social Media during the COVID-19 Pandemic*

Generation Y (so called "digital natives" and "millennials") represent the first generation that has spent its entire life in the digital environment, its life and work is profoundly affected by information technology [29] and it has experienced long periods of economic

prosperity and a rapid advance in instant communication technologies, social networking, and globalization [30]. These aspects have shaped Generation Y and influenced their social media use and buying behavior [29]. When comparing Generation X and Y, some authors, for example Bento et al. [31], assume that the interaction with social media seems to be more natural and intuitive for representatives of Generation Y than for representatives of Generation X. Their research revealed that Generation Y members consumed more content on Facebook brands' pages than Generation X. Generation Z (so called "iGeneration" and "post-millennials") uses technology extensively—on average, it uses up to five devices, compared to three for the millennials: smartphones, desktops, tablets, notebooks, TVs, tablets, and iPods [32]. Since the representatives of this generational cohort use smartphones basically for fun, they are sometimes called "the gamers" and "the supersmartphoners" [33]. Criteo Global App Survey [20] revealed significant differences in consumer preferences between generations during the first lockdown (first wave of the pandemic) in Italy—the youngest generations (Y and Z) focused on applications aimed at increasing and maintaining physical activity, mobile games, and streaming services, mainly related to leisure time. The older Generation X (1965–1980) showed an increased interest in applications intended for video conferencing (Zoom, WhatsApp, Skype), which was mainly related to the expanding trend of home offices. The biggest differences between the respondents were found in the social media Instagram and TikTok, where increased interest in these media was reported by up to 45% of the youngest consumers (Gen Y and Z) and only by 33% of older generation consumers (Gen X). Similarly, podcast- and music-oriented applications have become more popular among the younger generations. Regarding online shopping, there was an increased interest in online shopping applications among all generations, but representatives of Generations Y and Z had increased interest in food delivery, while the older Generation X preferred food and alcohol delivery (during the first lockdown). Thus, it is expected that the use of social media by representatives of different generations during the first wave of the COVID-19 pandemic will vary. This assumption led us to the formulation of the third hypothesis:

**Hypothesis 3 (H3).** *There are statistically significant generational differences (i.e., differences between selected generations of respondents) in the use of social media during the first wave of the COVID-19 pandemic (lockdown).*

*The Impact of Social Media on the Preference for e-Shops Advertized on Social Media*

Currently, most companies and organizations are relying on digital advertising and marketing techniques because online marketing still appears to be effective and efficient when compared with other forms of advertising and marketing [34]. Online communication through social media is one of the most used and useful tools for product promotion, since it is relatively inexpensive and enables organizations not only to convey a message for customers/clients but also interaction with their stakeholders [35]. Communication with modern technology increases the likelihood of purchasing, therefore it make sense to present products on social networks [36]. Korenkova et al. [37] in their research (primarily focused on consumers perception of different types of advertising in Slovakia) find that social media is the most influential advertisement among 21 types of advertisement. The research has also shown the difference between the older and younger generations (Y and Z) in perceiving all types of internet advertising. The older generation (Gen X) perceive the adverts as overloading (e.g., pop-ups on the web) to a greater extent than younger generations. This also applies to social media and other types of online advertising. This may be due to the fact that younger people who spend more time online are more accustomed to advertising. During the COVID-19 pandemic, there were many restrictions on opening hours or the complete closure of retails (especially during the lockdown), which led to an intensification of online shopping. According to the portal Criteo [20], 44% of Italian consumers (31% of them from generation Y and Z) downloaded a shopping application to their mobile phone during the lockdown, which was promoted through

social advertising (26%), television advertising (16%), or was suggested when using another application (15%). In this context, we investigated whether the respondents prefer shopping through e-shops and stores, which were designed for them by the social network as part of advertising or which were promoted on TV, radio, or on the web (in the period after the outbreak of the COVID-19 pandemic and the subsequent introduction of a restriction on opening hours complete closure of stores). Based on the above, two research hypotheses were formulated, namely:

**Hypothesis 4 (H4).** *There is a statistically significant link between the social media used (in terms of the type of network and the intensity of its use) and the preference for shopping through e-shops designed by a social network within advertising.*

*The Impact of Social Media on the Preference for e-Shops Promoted on Another Platforms*

In order for companies to be successful, competitive, and "visible", it is necessary to choose a combination of marketing communication activities to promote their products in order to attract the widest possible range of potential customers. The current customer is demanding, and competition in every area of sales is great, which means that everything depends on the attractiveness of the promotion of the companies. Customers buy and view products or services anywhere and anytime, that is why there needs to be available, immediate and accurate information so that their questions can be answered immediately. It is necessary to provide customer support 24/7. These facts contribute to the increasing popularity of advertising on social media and the increasing use of social media within the advertising strategy [37]. Several studies have been devoted to marketing communication through social media. For example, Duffet [38] carried out research on the YouTube marketing communication effect on attitudes of the generation Z consumers and found out that YouTube marketing communication has a favorable effect on the purchase intentions of the Generation Z cohort. The importance of advertising is also underlined by the study of Smith [39] which revealed that viewers who complete TrueView ads (i.e., watched until the end or at least 30 s) were 23 times more likely to visit or subscribe to a brand channel, watch more by that brand, or share the brand video. Even those viewers who do not watch to completion but are exposed to TrueView ads are still 10 times more likely to take one of those actions. In addition, when brands use TrueView, they see views of previously existing content increase by up to 500% after posting new videos. In the context of the above, the following hypothesis has been formulated:

**Hypothesis 5 (H5).** *There is a statistically significant link between the social media used (in terms of the type of network and the intensity of its use) and the preference for shopping through e-shops promoted on TV, radio, or on the web.*

## 3. Materials and Methods

Our research was carried out in two countries—Italy and Slovakia. The choice of these two countries was determined by previous research on consumer behavior conducted by the authors in the period before the outbreak of the COVID-19 pandemic results, which were published in indexed journals. In addition, these two countries were in a diametrically different situation during the first wave of the pandemic—while Italy (especially some regions) was one of the epicenters of the first coronavirus wave, the course of the first wave in Slovakia was very mild in contrast to many EU countries. At the end of May 2020 (i.e., at the approximate end of the first wave of the pandemic), 27 deaths were recorded as a result of COVID-19 in Slovakia and 33,415 in Italy, indicating a diametrically different situation in the two countries examined. In our cross-country research the method of back translation was used during questionnaire finalization. The original draft of the questionnaire was done in the Slovak language and it reflected Slovak habits and attitudes towards online purchasing and the most popular social media used in the time of the first lockdown of the country (March 2020). The increasing popularity of social media as well as the growing

number of calls on Facebook, Skype, and WhatsApp was driven partially by the change in working habits due to the lockdown. Although it has not been proven that the usage of applications based on communication like Zoom, MS Teams, and Slack impacts online shopping tendencies of their users, it is very likely that in connection with the compulsory use of social media due to the work duties, respondents also used other social networks and viewed the new content more often. The Slovak draft of the questionnaire was translated by an Italian native speaker lecturer and sent to Italy with a purpose of cooperation. After reviews from the Italian part, the questionnaire was enlarged with other questions reflecting the online shopping attitudes of Italians due to the pandemic situation of COVID-19 at that time. The Italian version of the questionnaire was subsequently translated back into Slovak by the lecturer, while the differences in translation could be considered synonymous and were negligible in translation. The completed questionnaire was reviewed by other Italian native speakers and lecturers and was ready to use. We did not observe any problems with understanding questions during the pre-testing stage.

The questionnaire was distributed online to email addresses. From a methodological point of view, the selection of respondents was made on the basis of the availability and voluntariness of the participants. Approximately 15,000 participants were contacted, while the total number of completed questionnaires reached the level of 937, with 336 respondents from Slovakia and 601 from Italy. The rate of return of the questionnaire was approximately 6%.

Data collection was performed via the questionnaire method during and immediately after the end of the first wave of the COVID-19 pandemic, i.e., in the period from February to June 2020. A validated questionnaire was used to collect data from respondents. The self-administered questionnaire was created in two language versions—Slovak and Italian. The initial identification and categorization questions were focused on the basic characteristics of the respondents—age, gender, education, residence, nationality, and number of household members. The questionnaire was focused on the consumer behavior of respondents during the first wave of the COVID-19 pandemic. Statistical software Statistica, Gretl, and MS Excel were used to process the obtained data.

Individual variables were subjected to a data normality test (Shapiro–Wilk test), which showed that none of the variables had a normal distribution. To verify individual hypotheses, validated methods such as Spearman's rank correlation coefficient were used to determine statistically significant relationships between the social media used and the preference for shopping through certain e-shops (Hypotheses 4 and 5).

Spearman's correlation coefficient, calculated according to Formula (1) follows [40]:

$$r_s = 1 - \frac{6\sum_{i=1}^{n} d^2}{n(n^2 - 1)},\tag{1}$$

where $d$ is the difference between ranks for the paired observations and $n$ is the number of paired observations.

The significance of rank correlation was tested using test statistics in the form of (2)

$$t = \frac{r_s}{\sqrt{(1 - r_s^2)/(n - 2)}}\tag{2}$$

The Mann–Whitney $U$ test is a nonparametric test based on comparing the medians of two independent samples. The test assumes the same variations in the two populations from which the two samples are compared, i.e., it tests whether the difference between the means ranks of the two groups is statistically significant or only random [41]. The calculation of the test statistic $U$ is performed by the following expression, while two different values will be obtained from the equation ($U_1$ and $U_2$)

$$U_{1(2)} = R_{1(2)} - \frac{n_{1(2)}\left(n_{1(2)} + 1\right)}{2}\tag{3}$$

where:

$n_{1(2)}$ is the sample size of the first (second) set and
$R_{1(2)}$ is the sum of the order in sample 1 (2).

The Kruskal–Wallis test is a nonparametric rank-sum statistical test which serves to test the null hypothesis that $k$ independent random samples come from identical populations against the alternative hypothesis that the medians of these populations are not all equal [42]. We used the following formula to calculate the Kruskal–Wallis $H$ test:

$$H = \frac{12}{N(N+1)} \sum_{i=1}^{k} \frac{R_i^2}{n_i} - 3(N+1), \tag{4}$$

where:

$R_i$ is the sum of ranks in the $i$th sample,
$n_i$ is the number of values contained in the $i$th sample, and
$N$ is the total number of observations in all samples combined.

## 4. Results

Primary data for the study were obtained from a sample of 937 respondents. The portfolio of the research sample consisted of 625 (66.7%) women and 312 (33.3%) men. Table 1 shows the distribution of the research sample by country and generation group of respondents, with respondents from Generation X (born in 1965–1979), Generation Y (born in 1980–1995), and Generation Z (born in 1996–2010).

**Table 1.** Survey sample composition.

| Respondents | Gen X | Gen Y | Gen Z | Total |
|---|---|---|---|---|
| Slovakia (SK) | 16 | 45 | 275 | 336 (35.9%) |
| Italy (IT) | 153 | 173 | 275 | 601 (64.1%) |
| Total | 169 (18%) | 218 (23.3%) | 550 (58.7%) | 937 |

Table 2 presents descriptive statistics for the use of social media during lockdown, where "Min" indicates the minimum value obtained in the questionnaire survey, "Max" the maximum value, std. dev. represents the standard deviation, Q1 lower quartile, Q3 upper quartile.

**Table 2.** Descriptive statistics on the use of social media during the first wave of the pandemic and lockdown.

| Social Media | Average | Median | Min | Max | Q1 | Q3 | Rank |
|---|---|---|---|---|---|---|---|
| Facebook | 3.4771 | 4.0 | 1.0 | 5.0 | 3.0 | 5.0 | 1.3789 |
| Twitter | 1.4056 | 1.0 | 1.0 | 5.0 | 1.0 | 1.0 | 0.9484 |
| Instagram | 3.6169 | 4.0 | 1.0 | 5.0 | 2.0 | 5.0 | 1.5564 |
| TikTok | 1.6638 | 1.0 | 1.0 | 5.0 | 1.0 | 2.0 | 1.2417 |
| YouTube | 3.6809 | 4.0 | 1.0 | 5.0 | 3.0 | 5.0 | 1.1756 |
| LinkedIn | 1.4386 | 1.0 | 1.0 | 5.0 | 1.0 | 1.0 | 0.9667 |
| WhatsApp | 3.6798 | 5.0 | 1.0 | 5.0 | 2.0 | 5.0 | 1.6218 |
| Messenger | 2.8954 | 3.0 | 1.0 | 5.0 | 1.0 | 4.0 | 1.5365 |
| Viber | 1.4386 | 1.0 | 1.0 | 5.0 | 1.0 | 1.0 | 1.0725 |
| Snapchat | 1.2337 | 1.0 | 1.0 | 5.0 | 1.0 | 1.0 | 0.7538 |
| Skype | 2.0832 | 1.0 | 1.0 | 5.0 | 1.0 | 3.0 | 1.2866 |
| Pinterest | 1.5955 | 1.0 | 1.0 | 5.0 | 1.0 | 2.0 | 1.1150 |
| Zoom | 1.8570 | 1.0 | 1.0 | 5.0 | 1.0 | 3.0 | 1.2146 |
| MS Teams | 1.9637 | 1.0 | 1.0 | 5.0 | 1.0 | 3.0 | 1.4153 |
| Slack | 1.1195 | 1.0 | 1.0 | 5.0 | 1.0 | 1.0 | 0.5779 |

Among the most commonly used social media (regardless of respondents' gender, education, age, and country of origin) was YouTube (in the first place), followed by WhatsApp, Instagram, Facebook, and Messenger. Other social media are used rarely or not at all. Regarding the use of social media, the highest variability was in the use of WhatsApp, Instagram, Messenger, MS Teams, and Facebook. By contrast, the lowest variability was in the use of Viber, LinkedIn, Twitter, Snapchat, and Slack.

The first research hypothesis (H1) is related to statistically significant differences between Italian and Slovak consumers in the use of social media (in terms of the type of network and the intensity of its use) during the first wave of the COVID-19 pandemic (lockdown). A 5-point frequency scale (ranging from never to very often) was used to determine the intensity of the use of social media. To verify the first hypothesis (H1), the Mann–Whitney $U$ test was used, the results of which are shown in Table 3.

**Table 3.** Mann–Whitney $U$ test (results of Hypothesis H1).

| Dependent Variable: Use of Social Media during the First Wave of the COVID-19 Pandemic | | Independent Variable: Country Market Tests Are Significant at $p < 0.050$ | | | | | | |
|---|---|---|---|---|---|---|---|---|
| | | Valid N | Rank Sum Group | U | Z | *p*-Value | Z Adj. | *p*-Value |
| Facebook | Italy | 601 | 260,156.0 | 79,255.00 | −5.4650 | 0.0000 | −5.6345 | 0.0000 |
| | Slovakia | 336 | 179,297.0 | | | | | |
| Twitter | Italy | 601 | 287,722.0 | 95,115.00 | 1.4731 | 0.1407 | 2.1440 | 0.0320 |
| | Slovakia | 336 | 151,731.0 | | | | | |
| Instagram | Italy | 601 | 272,102.0 | 91,201.00 | −2.4582 | 0.0140 | −2.5877 | 0.0097 |
| | Slovakia | 336 | 167,351.0 | | | | | |
| TikTok | Italy | 601 | 269,389.5 | 88,488.50 | −3.1410 | 0.0017 | −4.0170 | 0.0001 |
| | Slovakia | 336 | 170,063.5 | | | | | |
| YouTube | Italy | 601 | 258,404.0 | 77,503.00 | −5.9060 | 0.0000 | −6.1252 | 0.0000 |
| | Slovakia | 336 | 181,049.0 | | | | | |
| LinkedIn | Italy | 601 | 295,623.5 | 87,213.50 | 3.4619 | 0.0005 | 4.8544 | 0.0000 |
| | Slovakia | 336 | 143,829.5 | | | | | |
| WhatsApp | Italy | 601 | 366,801.5 | 16,035.50 | 21.3773 | 0.0000 | 23.0654 | 0.0000 |
| | Slovakia | 336 | 72,651.5 | | | | | |
| Messenger | Italy | 601 | 216,425.0 | 35,524.00 | −16.4721 | 0.0000 | −16.8822 | 0.0000 |
| | Slovakia | 336 | 223,028.0 | | | | | |
| Viber | Italy | 601 | 239,652.5 | 58,751.50 | −10.6257 | 0.0000 | −16.3408 | 0.0000 |
| | Slovakia | 336 | 199,800.5 | | | | | |
| Snapchat | Italy | 601 | 263,348.5 | 82,447.50 | −4.6615 | 0.0000 | −8.4784 | 0.0000 |
| | Slovakia | 336 | 176,104.5 | | | | | |
| Skype | Italy | 601 | 321,474.5 | 61,362.50 | 9.9686 | 0.0000 | 10.7524 | 0.0000 |
| | Slovakia | 336 | 117,978.5 | | | | | |
| Pinterest | Italy | 601 | 261,457.0 | 80,556.00 | −5.1376 | 0.0000 | −6.6467 | 0.0000 |
| | Slovakia | 336 | 177,996.0 | | | | | |
| Zoom | Italy | 601 | 307,447.0 | 75,390.00 | 6.4378 | 0.0000 | 7.2905 | 0.0000 |
| | Slovakia | 336 | 132,006.0 | | | | | |
| MS Teams | Italy | 601 | 238,557.5 | 57,656.50 | −10.9014 | 0.0000 | −12.5916 | 0.0000 |
| | Slovakia | 336 | 200,895.5 | | | | | |
| Slack | Italy | 601 | 276,188.0 | 95,287.00 | −1.4298 | 0.1528 | −3.6712 | 0.0002 |
| | Slovakia | 336 | 163,265.0 | | | | | |

The results of the Mann–Whitney $U$-test show that for all types of social media there are statistically significant differences in their use during first wave of the pandemic and lockdown between Italian and Slovak respondents. Hypothesis 1 can thus be confirmed. Descriptive statistics on the use of social media by Italian and Slovak respondents (Table 4) allow a clearer comparison, where std. dev. represents the standard deviation, Q1 lower quartile, Q3 upper quartile. The last column of the table shows the average ranking according to the intensity of use of a given social network in both countries.

**Table 4.** The use of social media by Italian and Slovak respondents.

| Social Media | Average Rating | | Median | | Q1 | | Q3 | | Std. Deviation | | Rank | |
|---|---|---|---|---|---|---|---|---|---|---|---|---|
| | IT | SK | IT | SK | IT | SK | IT | SK | IT | SK | IT | SK |
| Facebook | 3.2812 | 3.8274 | 4.0 | 4.0 | 2.0 | 3.0 | 4.0 | 5.0 | 1.4303 | 1.2069 | 4 | 3 |
| Twitter | 1.4725 | 1.2857 | 1.0 | 1.0 | 1.0 | 1.0 | 1.0 | 1.0 | 1.0342 | 0.7582 | 11 | 13 |
| Instagram | 3.5158 | 3.7977 | 4.0 | 4.0 | 2.0 | 3.0 | 5.0 | 5.0 | 1.5927 | 1.4744 | 2 | 4 |
| TikTok | 1.5408 | 1.8839 | 1.0 | 1.0 | 1.0 | 1.0 | 1.0 | 2.5 | 1.1234 | 1.4041 | 9 | 9 |
| Youtube | 3.5108 | 3.9851 | 4.0 | 4.0 | 3.0 | 3.0 | 4.0 | 5.0 | 1.1918 | 1.0830 | 3 | 2 |
| LinkedIn | 1.5541 | 1.2321 | 1.0 | 1.0 | 1.0 | 1.0 | 2.0 | 1.0 | 1.0728 | 0.6952 | 8 | 14 |
| WhatsApp | 4.6439 | 1.9554 | 5.0 | 1.0 | 4.0 | 1.0 | 5.0 | 3.0 | 0.6926 | 1.3564 | 1 | 7 |
| Messenger | 2.2596 | 4.0327 | 2.0 | 4.0 | 1.0 | 4.0 | 3.0 | 5.0 | 1.3314 | 1.1828 | 6 | 1 |
| Viber | 1.0399 | 2.1518 | 1.0 | 1.0 | 1.0 | 1.0 | 1.0 | 3.0 | 0.3292 | 1.4915 | 15 | 6 |
| Snapchat | 1.0815 | 1.5060 | 1.0 | 1.0 | 1.0 | 1.0 | 1.0 | 1.0 | 0.4062 | 1.0845 | 13 | 11 |
| Skype | 2.4077 | 1.5030 | 2.0 | 1.0 | 1.0 | 1.0 | 4.0 | 2.0 | 1.3274 | 0.9712 | 5 | 12 |
| Pinterest | 1.4160 | 1.9167 | 1.0 | 1.0 | 1.0 | 1.0 | 1.0 | 3.0 | 0.9452 | 1.3086 | 12 | 8 |
| Zoom | 2.0466 | 1.5179 | 1.0 | 1.0 | 1.0 | 1.0 | 3.0 | 1.0 | 1.2601 | 1.0479 | 7 | 10 |
| MS Teams | 1.5408 | 2.7202 | 1.0 | 3.0 | 1.0 | 1.0 | 1.0 | 4.0 | 1.1279 | 1.5566 | 10 | 5 |
| Slack | 1.0732 | 1.2024 | 1.0 | 1.0 | 1.0 | 1.0 | 1.0 | 1.0 | 0.4669 | 0.7294 | 14 | 15 |

Our research revealed that during the first wave of the pandemic, the most used media in Italy were WhatsApp, Instagram, YouTube, Facebook, and Skype, and in Slovakia Messenger, YouTube, Facebook, Instagram, and MS Teams. The biggest differences (in frequency of use) between the two countries were in LinkedIn, WhatsApp, Messenger, Viber, Skype, Pinterest, and MS Teams (difference of more than 5 positions in the order). Hypothesis H1 is confirmed. The preference of WhatsApp by Italian respondents could emanate from the fact that it is a more intimate and confidential mean of communication, implying closer and more friendly relationships between people than other media and networks. Similarly, the Slovaks also prioritize a social network such as Messenger, which facilitates confidential exchanges between people who know each other well; they also make greater use of social media that allow personal visibility and convey individual brands, in line with the current narcissistic trend in society, where appearing has become an imperative for strengthening one's social identity. Probably, during the isolation imposed by the lockdown many people felt the need to share thoughts, images, and content with their network of acquaintances and followers, in order to feel alive and connected to their social communities.

The Mann–Whitney *U*-test was used to verify the assumption of statistically significant differences in the use of social media between men and women (hypothesis H2), the results of which are shown in Table 5. Table 6 shows descriptive statistics on the use of social media by men and women.

The results show that there are statistically significant gender differences in the use of Facebook, Twitter, Instagram, TikTok, LinkedIn, Messenger, Pinterest, MS Teams, and Slack. Most of these media (Facebook, Instagram, TikTok, Messenger, and Pinterest) are preferred by women (see also Table 6), while men are more likely to use Twitter, LinkedIn, MS Teams, and Slack. Hypothesis H2 is confirmed.

The Kruskal–Wallis *H* test was used to test the third hypothesis (H3), which assumes the existence of statistically significant differences in the use of social media during the first wave of the COVID-19 pandemic between the three generational groups of respondents—Gen X (aged 41 and over), Gen Y (aged 25–40 years), and Gen Z (under the 25).

**Table 5.** Mann–Whitney U Test (results of Hypothesis H2).

| Dependent Variable: Use of Social Network during the First Wave of the COVID-19 Pandemic | | Independent Variable: Gender Market Tests Are Significant at $p < 0.050$ | | | | | | |
|---|---|---|---|---|---|---|---|---|
| | | Valid N | Rank Sum Group | U | Z | *p*-Value | Z Adj. | *p*-Value |
| Facebook | Men | 312 | 134,730.0 | 85,902.00 | −2.9706 | 0.0030 | −3.06265 | 0.0022 |
| | Women | 625 | 304,723.0 | | | | | |
| Twitter | Men | 312 | 158,094.5 | 85,733.50 | 3.0137 | 0.0026 | 4.38637 | 0.0000 |
| | Women | 625 | 281,358.5 | | | | | |
| Instagram | Men | 312 | 120,161.0 | 71,333.00 | −6.7022 | 0.0000 | −7.05525 | 0.0000 |
| | Women | 625 | 319,292.0 | | | | | |
| TikTok | Men | 312 | 131,185.0 | 82,357.00 | −3.8786 | 0.0001 | −4.96026 | 0.0000 |
| | Women | 625 | 308,268.0 | | | | | |
| YouTube | Men | 312 | 143,284.0 | 94,456.00 | −0.7796 | 0.4357 | −0.80849 | 0.4188 |
| | Women | 625 | 296,169.0 | | | | | |
| LinkedIn | Men | 312 | 164,208.5 | 79,619.50 | 4.5797 | 0.0000 | 6.42191 | 0.0000 |
| | Women | 625 | 275,244.5 | | | | | |
| WhatsApp | Men | 312 | 142,033.5 | 93,205.50 | −1.0999 | 0.2714 | −1.18670 | 0.2353 |
| | Women | 625 | 297,419.5 | | | | | |
| Messenger | Men | 312 | 134,427.0 | 85,599.00 | −3.0482 | 0.0023 | −3.12405 | 0.0018 |
| | Women | 625 | 305,026.0 | | | | | |
| Viber | Men | 312 | 144,621.0 | 95,793.00 | −0.4371 | 0.6620 | −0.67219 | 0.5015 |
| | Women | 625 | 294,832.0 | | | | | |
| Snapchat | Men | 312 | 144,592.0 | 95,764.00 | −0.4445 | 0.6567 | −0.80851 | 0.4188 |
| | Women | 625 | 294,861.0 | | | | | |
| Skype | Men | 312 | 145,027.0 | 96,199.00 | −0.3331 | 0.7391 | −0.35930 | 0.7194 |
| | Women | 625 | 294,426.0 | | | | | |
| Pinterest | Men | 312 | 124,396.0 | 75,568.00 | −5.6175 | 0.0000 | −7.26756 | 0.0000 |
| | Women | 625 | 315,057.0 | | | | | |
| Zoom | Men | 312 | 146,590.0 | 97,238.00 | 0.0670 | 0.9466 | 0.07585 | 0.9395 |
| | Women | 625 | 292,863.0 | | | | | |
| MS Teams | Men | 312 | 155,584.5 | 88,243.50 | 2.3708 | 0.0178 | 2.73840 | 0.0062 |
| | Women | 625 | 283,868.5 | | | | | |
| Slack | Men | 312 | 153,985.5 | 89,842.50 | 1.9612 | 0.0499 | 5.03578 | 0.0000 |
| | Women | 625 | 285,467.5 | | | | | |

**Table 6.** The use of social media by men and women.

| Social Media | Average Rating | | Median | | Q1 | | Q3 | | Std. Deviation | |
|---|---|---|---|---|---|---|---|---|---|---|
| | Men | Women | Men | Women | Men | Women | Men | Women | Men | Women |
| Facebook | 3.2692 | 3.5808 | 4.0 | 4.0 | 2.0 | 3.0 | 4.8 | 5.0 | 1.4450 | 1.3338 |
| Twitter | 1.5224 | 1.3472 | 1.0 | 1.0 | 1.0 | 1.0 | 4.5 | 1.0 | 0.9654 | 0.9351 |
| Instagram | 3.1282 | 3.8608 | 4.0 | 4.0 | 1.0 | 3.0 | 2.0 | 5.0 | 1.6007 | 1.4757 |
| TikTok | 1.3974 | 1.7968 | 1.0 | 1.0 | 1.0 | 1.0 | 5.0 | 2.0 | 0.9963 | 1.3284 |
| Youtube | 3.6154 | 3.7136 | 4.0 | 4.0 | 3.0 | 3.0 | 1.0 | 5.0 | 1.2499 | 1.1363 |
| LinkedIn | 1.7147 | 1.3008 | 1.0 | 1.0 | 1.0 | 1.0 | 5.0 | 1.0 | 1.1753 | 0.8100 |
| Whatsapp | 3.6346 | 3.7024 | 4.0 | 5.0 | 2.0 | 2.0 | 2.0 | 5.0 | 1.5636 | 1.6509 |
| Messenger | 2.6731 | 3.0064 | 3.0 | 3.0 | 1.0 | 1.0 | 5.0 | 5.0 | 1.4597 | 1.5628 |
| Viber | 1.3686 | 1.4736 | 1.0 | 1.0 | 1.0 | 1.0 | 4.0 | 1.0 | 0.9533 | 1.1265 |
| Snapchat | 1.1827 | 1.2592 | 1.0 | 1.0 | 1.0 | 1.0 | 1.0 | 1.0 | 0.6070 | 0.8165 |
| Skype | 2.0385 | 2.1056 | 2.0 | 1.0 | 1.0 | 1.0 | 1.0 | 3.0 | 1.2340 | 1.3126 |
| Pinterest | 1.2179 | 1.7840 | 1.0 | 1.0 | 1.0 | 1.0 | 3.0 | 3.0 | 0.6785 | 1.2364 |
| Zoom | 1.8205 | 1.8752 | 1.0 | 1.0 | 1.0 | 1.0 | 1.0 | 3.0 | 1.1255 | 1.2572 |
| MS Teams | 2.1090 | 1.8912 | 1.0 | 1.0 | 1.0 | 1.0 | 3.0 | 3.0 | 1.4593 | 1.3883 |
| Slack | 1.2372 | 1.0608 | 1.0 | 1.0 | 1.0 | 1.0 | 3.0 | 1.0 | 0.8028 | 0.4116 |

The results of the testing of hypothesis H3 (Table 7) confirm the existence of statistically significant differences between the selected generations of respondents in almost all cases of use of social media (except the social network Twitter and Slack). Hypothesis H3 is confirmed. By using multiple comparisons of the mean ranks for all groups, we determined that there are differences between several groups, the main difference being mostly between Generations X and Z (except for the use of MS Teams) and between Generations Y and Z (except for the use of Facebook, Viber, Skype, Pinterest, and MS Teams). Differences between Generation X and Y are the least common, while differences between these generation groups of respondents were demonstrated only in the case of using Facebook, Instagram, YouTube, Messenger, Skype, and MS Teams. Table 8 presents descriptive statistics on the use of individual social media by Generations X, Y, and Z.

**Table 7.** Kruskal–Wallis *H* test (results of hypothesis H3).

| Use of Social Media during the Pandemic | Generation X (Mean Rank) | Generation Y (Mean Rank) | Generation Z (Mean Rank) | Kruskal-Wallis Test: H | *p*-Value |
|---|---|---|---|---|---|
| Facebook | 404.8343 | 496.0000 | 478.0145 | 13.0535 | 0.0015 *** |
| Twitter | 482.3314 | 468.6307 | 465.0500 | 1.1178 | 0.5718 |
| Instagram | 201.4734 | 419.6606 | 570.7600 | 277.1966 | 0.0000 *** |
| TikTok | 359.8846 | 419.4335 | 522.1745 | 91.6216 | 0.0000 *** |
| Youtube | 289.2959 | 451.1927 | 531.2764 | 112.4911 | 0.0000 *** |
| LinkedIn | 563.4615 | 530.7248 | 415.5091 | 105.0307 | 0.0000 *** |
| Whatsapp | 519.0444 | 539.2431 | 425.7809 | 40.1537 | 0.0000 *** |
| Messenger | 362.4675 | 452.8165 | 508.1491 | 40.4159 | 0.0000 *** |
| Viber | 418.4704 | 450.3486 | 491.9191 | 25.7104 | 0.0000 *** |
| Snapchat | 429.5947 | 439.2546 | 492.8982 | 34.7521 | 0.0000 *** |
| Skype | 559.4852 | 475.0963 | 438.7800 | 30.0875 | 0.0000 *** |
| Pinterest | 421.9852 | 457.2936 | 488.0864 | 13.7982 | 0.0010 *** |
| Zoom | 548.2811 | 490.7408 | 436.0218 | 30.87740 | 0.0000 *** |
| MTeams | 504.9970 | 424.2317 | 475.6836 | 12.3952 | 0.0020 *** |
| Slack | 474.1775 | 467.6193 | 467.9564 | 0.4991 | 0.7791 |

*** statistical significance at 1%.

**Table 8.** The use of social media by Generations X, Y, and Z.

| Social Media | Average Rating | | | Median | | | Q1 | | | Q3 | | | Std. Deviation | | |
|---|---|---|---|---|---|---|---|---|---|---|---|---|---|---|---|
| | X | Y | Z | X | Y | Z | X | Y | Z | X | Y | Z | X | Y | Z |
| Facebook | 3.0828 | 3.6376 | 3.5345 | 4.0 | 4.0 | 4.0 | 1.0 | 3.0 | 3.0 | 4.0 | 5.0 | 5.0 | 1.5562 | 1.2851 | 1.3366 |
| Twitter | 1.4201 | 1.4220 | 1.3945 | 1.0 | 1.0 | 1.0 | 1.0 | 1.0 | 1.0 | 1.0 | 1.0 | 1.0 | 0.8632 | 0.9724 | 0.9651 |
| Instagram | 1.8935 | 3.3578 | 4.2491 | 1.0 | 4.0 | 5.0 | 1.0 | 2.0 | 4.0 | 3.0 | 5.0 | 5.0 | 1.2538 | 1.5747 | 1.1522 |
| TikTok | 1.0769 | 1.3349 | 1.9745 | 1.0 | 1.0 | 1.0 | 1.0 | 1.0 | 1.0 | 1.0 | 1.0 | 3.0 | 0.4082 | 0.8162 | 1.4383 |
| Youtube | 2.8402 | 3.6239 | 3.9618 | 3.0 | 4.0 | 4.0 | 2.0 | 3.0 | 3.0 | 4.0 | 4.0 | 5.0 | 1.1819 | 1.1261 | 1.0629 |
| LinkedIn | 1.8462 | 1.7615 | 1.1855 | 1.0 | 1.0 | 1.0 | 1.0 | 1.0 | 1.0 | 3.0 | 2.0 | 1.0 | 1.1902 | 1.2212 | 0.6431 |
| Whatsapp | 4.2308 | 4.1606 | 3.3200 | 4.0 | 5.0 | 4.0 | 4.0 | 4.0 | 1.0 | 5.0 | 5.0 | 5.0 | 1.0235 | 1.3152 | 1.7791 |
| Messenger | 2.2840 | 2.8028 | 3.1200 | 2.0 | 3.0 | 3.0 | 1.0 | 1.0 | 2.0 | 3.0 | 4.0 | 5.0 | 1.4192 | 1.4183 | 1.5635 |
| Viber | 1.1598 | 1.2890 | 1.5836 | 1.0 | 1.0 | 1.0 | 1.0 | 1.0 | 1.0 | 1.0 | 1.0 | 1.0 | 0.7016 | 0.8055 | 1.2249 |
| Snapchat | 1.0473 | 1.0826 | 1.3509 | 1.0 | 1.0 | 1.0 | 1.0 | 1.0 | 1.0 | 1.0 | 1.0 | 1.0 | 0.2847 | 0.3750 | 0.9245 |
| Skype | 2.5266 | 2.1009 | 1.9400 | 2.0 | 2.0 | 1.0 | 1.0 | 1.0 | 1.0 | 4.0 | 3.0 | 3.0 | 1.3321 | 1.2840 | 1.2436 |
| Pinterest | 1.3254 | 1.5321 | 1.7036 | 1.0 | 1.0 | 1.0 | 1.0 | 1.0 | 1.0 | 1.0 | 1.0 | 2.0 | 0.8132 | 1.0435 | 1.2046 |
| Zoom | 2.1953 | 1.9587 | 1.7127 | 2.0 | 1.0 | 1.0 | 1.0 | 1.0 | 1.0 | 3.0 | 3.0 | 2.0 | 1.2501 | 1.2455 | 1.1677 |
| MS Teams | 2.1124 | 1.7110 | 2.0182 | 1.0 | 1.0 | 1.0 | 1.0 | 1.0 | 1.0 | 3.0 | 2.0 | 3.0 | 1.4118 | 1.2678 | 1.4610 |
| Slack | 1.1243 | 1.1055 | 1.1236 | 1.0 | 1.0 | 1.0 | 1.0 | 1.0 | 1.0 | 1.0 | 1.0 | 1.0 | 0.5794 | 0.5106 | 0.6030 |

Statistical analysis showed that Generation X uses media such as LinkedIn, WhatsApp, Zoom, MS Teams, and Skype more often than two younger generations, which may be due to the fact that they represent either "older" (e.g., Skype) or more professional (LinkedIn, MS Teams) media/networks. Other social media is mostly used by the youngest Generation Z (Instagram, TikTok, YouTube, Messenger, Viber, Snapchat, and Pinterest). Generation Y is

somewhere in the imaginary middle between these two generations, with higher similarity being found with Generation X. No differences in the use of Facebook, Viber, or Skype were found between Generation Y and Generation Z.

The aim of testing the last two hypotheses was to confirm or reject the claim of the existence of a statistically significant relationship between the use of individual social media and the preference for shopping through e-shops designed by the social network within advertising or otherwise promoted (on TV, radio, on the web). One of the aims of the research was to find out which social media has a stronger "impact" on the consumer and his/her decision to use the e-shop advertised by the social network, i.e., whether an advertisement for an e-shop offered by a certain social network more likely lead to the consumer's decision to use the offered e-shop for his purchase (H4). Spearman's correlation coefficient was used for hypothesis testing (Table 9). The indicated values of the Spearman rank correlation coefficient are significant at the level of $p < 0.050$.

The research has shown that although there are several links between the social media used and between the use of social media and the preference for shopping through e-shops promoted on social media, TV, radio, or the web, the values of these correlations are usually low. We can conclude that there is very little connection between the use of different social media, i.e., using one social network is not related to using another social network (for example, if someone uses Facebook, it does not mean that he/she also uses Instagram). Significant correlations are observable in two media—Facebook and Messenger (the correlation coefficient has a value of 0.5775), which we consider meaningful, given that they are often perceived as one network, because they are interconnected. On the contrary, the platform WhatsApp shows a negative correlation with some social media and applications (e.g., Messenger, Viber, MS Teams). Between the variables (1) social media and (2) preference for shopping through e-shops and stores, which a) were promoted on a social network, or b) were promoted on TV, radio, or on the web, there is a very low correlation. However, there is a relatively strong correlation between the variables expressing the preference for shopping through e-shops and stores that were promoted on the social network and those that were advertised on TV, radio, or on the web.

**Table 9.** Spearman correlation coefficient (results of hypothesis H4 and H5).

| | Facebook | Twitter | Instagram | TikTok | Youtube | LinkedIn | WhatsApp | Messenger | Viber | Snapchat | Skype | Pinterest | Zoom | MS Teams | Slack | Preference of e-Shop Advertised By Social Network | e-Shops Promoted on TV, Radio or on the Web |
|---|---|---|---|---|---|---|---|---|---|---|---|---|---|---|---|---|---|
| Facebook | 1.0000 | −0.0660 | 0.3039 | 0.0115 | 0.2007 | 0.0382 | −0.0359 | 0.5775 | 0.0146 | 0.0148 | −0.0030 | 0.0934 | −0.0068 | 0.0865 | 0.0022 | 0.0634 | 0.0609 |
| Twitter | | 1.0000 | 0.0316 | 0.1859 | −0.0117 | 0.2910 | 0.0811 | −0.0822 | 0.0718 | 0.2129 | 0.1513 | 0.0833 | 0.1306 | 0.0329 | 0.3052 | −0.0315 | −0.0086 |
| Instagram | | | 1.0000 | 0.3008 | 0.3646 | −0.1534 | 0.0858 | 0.2182 | 0.0672 | 0.1194 | −0.0057 | 0.1085 | −0.0093 | −0.0159 | −0.0438 | 0.2100 | 0.1007 |
| TikTok | | | | 1.0000 | 0.1227 | 0.0159 | −0.0244 | 0.0402 | 0.2498 | 0.3336 | 0.0182 | 0.1763 | 0.0473 | 0.0762 | 0.1462 | 0.1392 | 0.1200 |
| Youtube | | | | | 1.0000 | −0.1134 | −0.0675 | 0.2401 | 0.1359 | 0.0983 | −0.0847 | 0.1074 | −0.1032 | 0.0674 | −0.0458 | 0.0520 | 0.0849 |
| LinkedIn | | | | | | 1.0000 | 0.1451 | −0.0693 | 0.0304 | 0.1148 | 0.3064 | 0.0185 | 0.2427 | 0.1351 | 0.3151 | −0.0193 | −0.0225 |
| WhatsApp | | | | | | | 1.0000 | −0.3316 | −0.3188 | −0.1558 | 0.3393 | −0.1576 | 0.2746 | −0.2831 | −0.0757 | 0.0335 | −0.0783 |
| Messenger | | | | | | | | 1.0000 | 0.2181 | 0.1032 | −0.1499 | 0.1547 | −0.1432 | 0.2777 | 0.0197 | 0.0872 | 0.1376 |
| Viber | | | | | | | | | 1.0000 | 0.3852 | −0.1025 | 0.2443 | −0.0574 | 0.2898 | 0.2354 | 0.0668 | 0.1758 |
| Snapchat | | | | | | | | | | 1.0000 | 0.0533 | 0.2384 | 0.0447 | 0.2090 | 0.3229 | 0.1335 | 0.1920 |
| Skype | | | | | | | | | | | 1.0000 | 0.0049 | 0.3796 | −0.0843 | 0.1129 | 0.0461 | −0.0200 |
| Pinterest | | | | | | | | | | | | 1.0000 | 0.0414 | 0.1342 | 0.1448 | 0.1054 | 0.1130 |
| Zoom | | | | | | | | | | | | | 1.0000 | 0.0094 | 0.1846 | 0.0428 | −0.0133 |
| MTeams | | | | | | | | | | | | | | 1.0000 | 0.1792 | 0.0201 | 0.1589 |
| Slack | | | | | | | | | | | | | | | 1.0000 | 0.1306 | 0.1706 |
| preference of e-shop advertised by network | | | | | | | | | | | | | | | | 1.0000 | 0.6717 |
| e-shops promoted on TV, radio or on the web | | | | | | | | | | | | | | | | | 1.0000 |

## 5. Discussion

Appel et al. [16] argue that "it is important to consider the future of social media in the context of consumer behavior and marketing, since social media has become a vital marketing and communications channel for businesses, organizations and institutions alike" (p. 79). Lv et al. [8] believe that "the current COVID-19 global epidemic, the contactless shopping process and the continued global economic downturn have made the trend of social e-commerce more obvious and inevitable" (p. 14) and the combination of social media and e-commerce will be more efficient in the future. The lockdown and social distancing during the COVID-19 pandemic significantly changed consumer behavior and made consumption time-bound and location-bound. People not being able to purchase in stores and outlets resulted in stores having to "come" to consumers. It can be assumed that adaption of people to house arrest for a prolonged period of time will likely to lead to their adoption of newer technologies which facilitate work, study, and consumption in a more convenient manner [43].

One of the main findings of our research is the fact that the use of social media depends on several personality characteristics, as differences were found with respect to different identifiers of respondents, such as gender or age. The importance of knowing the preferences of individual groups of respondents is the basis of segmentation and targeted customer orientation. When considering the correct placement of an advertisement or promotion of a product or company, information on which age group prefers which social network is key, e.g., if a company wants to target its product for a generation over 40, choosing Instagram would probably not be the optimal choice because this network is not widely used by this generation group (Gen X).

It was found that Facebook, Messenger, Instagram, and YouTube are among the most popular and most used social media. Although differences between Italy and Slovakia were found in the order of these media (in terms of intensity of use), they are still popular in both groups. For other media (networks, platforms, and applications), the results are quite different. Thus, if the company, the retailer, is interested in reaching as many consumers/potential customers as possible, it would be appropriate to focus on these media. An interesting finding is the relatively low interest in the social network Twitter, which is very popular in the USA, for example. The most used social media (not only during the pandemic) are Facebook, Instagram, and YouTube which are most often used by young consumers (regardless of their country of origin). According the free online stats tool "Statcounter Global Stats" provided by Statcounter which maps social media usage, in the period March–May 2020, the most used network in Italy was Facebook, followed by Instagram, Twitter, Pinterest, and YouTube. In Slovakia, the following media were the most used in the given time period: Facebook, Pinterest, Instagram, Twitter, and Reddit. Differences in the final order of social media (in terms of intensity and frequency of their use) in Italy and Slovakia (as well as compared to the online statistical tool) may be due to the composition of the survey sample—the average age of Italian respondents was 32 years and Slovak respondents 23.5 years. In addition, seniors (representatives of the Baby Boomers and Silent Generation cohorts) who prefer different social media and platforms than the younger generations were not included in the research.

Although differences were found between respondents from both countries, as well as between men and women, it can be stated that the most significant differences were in terms of the age of the respondents. Differential analysis with respect to age groups has revealed that the most commonly used media is the same in all age categories, although the older generation tends to use "older" media (networks and applications), while the younger generation prefers the newer ones. At the same time, it was proven (and previously expected) that the use of social media is the dominant feature of Generation Z.

All hypotheses were confirmed (with small exceptions within some groups). The most surprising results were obtained by testing hypotheses 4 and 5—although correlations between variables have been demonstrated, these correlations were very low, especially when it came to variable relating to advertising on social media. Research revealed rela-

tively weak correlations between the use of social media and the preference for shopping through e-shops promoted on social media, TV, radio, or the web. However, a similar study by Poornima et al. [34] that focused on the examination of relationship between e-advertisement and consumer buying behavior among Generation Y and Z consumers showed that most of the young respondents were influenced by online advertisements, especially those on social media (65%). Furthermore, 89% of respondents indicated that e-advertisement increased their shopping trends whereupon the authors concluded that online advertising is the most important factor to predict consumer buying behavior. It is to be noted that the study was conducted before the outbreak of the pandemic which fundamentally changed peoples' preferences, attitudes, and behaviors. The study by Fondevila-Gascón et al. [44] was focused on the mobile manufacturers online advertising (Apple, Samsung, Xiaomi, and BQ) with emphasis on social media and its direct influence on consumer behavior. One of the confirmed assumptions of their study is that the older Generation X is influenced more by advertising of the product than younger Generation Y whereby no significant gender differences were observed.

However, in the context of understanding how the COVID-19 pandemic has affected online self-disclosures, Nabity-Grover et al. [17] point to an interesting and considerable fact that due to social pressures and fear of shaming, many consumer habits may be less apparent (especially in the cases of dining and travel) which makes social media a less reliable source of consumer data.

## 6. Conclusions

The literature about consumer behavior during COVID-19 pandemic has expanded significantly in the recent period. Present study offers an insight into the use of social media by Italian and Slovak consumers during the first wave of the COVID-19 pandemic (lockdown) and into the influencing mechanism of social media on consumer behavior. One of the main aims of this study is to use the knowledge gained to create a kind of "consumer profile" during the COVID-19 pandemic.

The results revealed the existence of statistically significant differences in the use of all types of social media during the first wave of the COVID-19 between Italian and Slovak consumers. However, despite a different propensity to use various social media, the study shows also that, during the pandemic, both Italian and Slovak consumers favored social media such as WhatsApp (Italians) and Messenger (Slovaks), which allow for the development of relationships intimate between people, probably seeking to maintain and consolidate bonds of friendship to govern the anxieties and fears connected to the dramatic experience of social distancing due to Covid 19. Moreover, the results highlight that there are gender differences in the use of social media. In particular, it emerges that women prefer social media more based on images and personal content like Facebook, Instagram, Pinterest, and TikTok, while males use more social media based on work and professional content like Twitter and LinkedIn. This difference could reflect aspects related to psychological profiles of different extroversion/introversion of the two observed groups, an opening to new research routes. Our study also shows that there are generational differences in the use of social media, which highlight how Generation X favor professional social channels connected to work needs, like LinkedIn, WhatsApp, Zoom, MS Teams, and Skype, compared to younger ones, more oriented towards social entertainment and leisure channels. With reference to the link between the use of social media and the propensity to make online purchases by respondents, the research reveals only weak relationships between the variables. The results may be considered surprising, also in light of the indications that other studies have highlighted (Barton et al., 2014; Pencarelli et al. 2020) [45,46]. Therefore, at the moment it is difficult to comment and formulate conclusions, in the absence of statistical evidence on the impact of social media on consumer purchasing behavior; however, the result brings out some preliminary theoretical implications. First of all, it is likely that some social media (Facebook, Messenger, Instagram, Twitter), due

to their characteristics, have a greater impact on consumer behavior than others (e.g., WhatsApp and LinkedIn), whose aims are different.

Basically, there are social media that are more oriented towards social commerce, directing web surfers to e-commerce sites, while some social media are mainly aimed at fostering friendly and intimate relationships (WhatsApp) or professional relationships (LinkedIn). Some social media are more suited to generating online word-of-mouth, sharing information about products, publishing opinions on their functionality, and thus influencing friends and users in their purchasing and consumption processes. The impact of social media on the propensity to purchase online is strengthened when communication via social channels is associated with communication with traditional media such as tv and radio, highlighting the importance from a theoretical and managerial perspective of an integrated use of communication processes by of businesses.

From a theoretical and practical point of view, the research also suggests the importance of differentiating social channels on the basis of the objectives of simple social communication compared to the narrower aims of social commerce. A differentiated approach to carrying out effective social campaigns is also appropriate when addressing consumers who are different from a geographical, gender, and generational point of view, underlining the strategic importance of refining the segmentation processes of demand. In line with these assumptions, the importance for companies to prepare a plan of social channels to be used according to the objectives sought for the various market segments to be reached and the contents to be conveyed emerges, which may depend on the needs to create and entertain individuals or communities, listening, entertaining and informing them, and promoting certain brands to influence online purchases.

Moreover, in terms of managerial implications, we can say that the study suggests that business managers should use social media to optimize the entire customer experience at all different stages of the purchasing cycle, starting from the problem analysis phase, then proceeding to information search, evaluation of alternatives, purchase, and post-purchase. For each of these stages, social media managers can choose the most effective social channel to allow potential buyers to take in the information they need to optimize their purchase decision, such as reviews, opinions from friends or experts, or influencers, encourage the use of e-commerce platforms and gather opinions on the post-purchase behavior. Particularly important is the ZMOT (zero moment of truth) or bones when consumers search for information online about a certain product before buying it. Managers therefore need to create sharing moments, recommendation indicators (such as a like on Facebook), recording of testimonial opinions, wish lists, user forums, etc. [47].

Despite the limit that every study can inevitably have, this study highlights the importance of avoiding deterministic automatisms between the use of social media and online purchasing processes, suggesting the need to learn, from a theoretical and managerial point of view, the importance of the set of digital marketing processes to truly influence consumer purchasing behavior, according to the route indicated by Kannan [48]. Given the lack of research on examining gender differences in the impact of social media on the decisions and attitudes when shopping online, we believe that this study may contribute to filling this gap.

**Author Contributions:** Conceptualization, V.A.T., T.P., and V.Š.; methodology, V.A.T., T.P., and V.Š.; software and validation, M.K.; formal analysis, M.K.; investigation, T.P. and V.Š.; resources, V.A.T. and V.Š.; data curation, T.P. and V.Š.; writing—original draft preparation, review, and editing, V.A.T., T.P., V.Š., and M.K.; visualization, V.A.T.; supervision, T.P. and R.F.; project administration and funding acquisition, R.F. All authors have read and agreed to the published version of the manuscript.

**Funding:** This research was funded by Vedecká Grantová Agentúra MŠVVaŠ SR a SAV (VEGA).

**Institutional Review Board Statement:** Not applicable.

**Informed Consent Statement:** Informed consent was obtained from all subjects involved in the study.

**Data Availability Statement:** Data available on request due to restrictions, e.g., privacy or ethical. The data presented in this study are available on request from the corresponding author. The data are not publicly available due to GDPR.

**Acknowledgments:** This research is one of the partial outputs under the scientific research grants VEGA 1/0694/20 "Relational marketing research—perception of e-commerce aspects and its impact on purchasing behavior and consumer preferences" and VEGA 1/0609/19 "Research on the development of electronic and mobile commerce in the aspect of the impact of modern technologies and mobile communication platforms on consumer behavior and consumer preferences".

**Conflicts of Interest:** The authors declare no conflict of interest.

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
