# Peer review of "The Use of Social Media and Its Impact on Shopping Behavior of Slovak and Italian Consumers during COVID-19 Pandemic"

_sustainability, doi:10.3390/su13041710_

Round 1

Reviewer 1 Report

Up-to-date topic, I appreciate it. The countries selection should be explained in a more detailed way. Additional information concerning the sample size limitations is missing, although explained, but in general. Did you apply the back translation method or some kind of pre-testing? The liaison between social media and consumer behaviour should be strengthened within the context of the paper title (especially when speaking about conclusions).

Author Response

Dear Reviewer, we are very thankful for giving us the opportunity to submit a revised draft of our manuscript titled "The use of social media and its impact on shopping behavior of Slovak and Italian consumers during COVID-19 pandemic" to Sustainability journal. We really appreciate the time and effort that have dedicated to providing your valuable feedback on our manuscript. We are grateful to for your insightful comments on our paper. We have been able to incorporate changes to reflect most of the suggestions provided by you.

Reviewer 2 Report

While, this paper might of interest to the journal of sustainability, in its current form it is quite weak. There are various aspects that the authors should consider:

Introduction- The first paragraph of this section from L26 to L 34 does not really connect to the next paragraph. It would have been much better if the authors could instead start how COVID 19 has further amplified the importance of social media and how it has been influencing consumer behaviours. It is suggested that the authors move line 105 to 106 in the introduction section instead of the materials and methods section. As it is now, the introduction section is quite weak. The authors need to argue why it is interesting from a theoretical perspective to study the link between social media and its impact on consumer behavior. What is the gap in the literature ? More effort should also be put in terms of highlighting how the study intend to contribute theoretically and practically to the literature.

L128-140: are the authors stating that there is only one study that has studied the relationship between use of social media and gender ? The authors need to rethink if it is logical to develop a hypothesis based on this study only.

When it comes to structure and for a clearer flow of the text and argument. It is suggested that the author shifts the text in methods(L107 to L113) to  L231 and not to mix it with the literature review section.

When it comes to data collection, it is unclear how the authors selected the respondents of the study. Were there any selection criteria applied ? How was the survey administered ? What was the response rate ?

it is also suggested the authors have clear sub-headings of the literature review and methods sections and not to combine them.

Discussion- this section is very weak. The authors need to put more emphasize in terms of how the study contributes to the current literature, meaning the theoretical and practical implication of this study needs to be highlighted.

Author Response

Dear Reviewer, we are very thankful for giving us the opportunity to submit a revised draft of our manuscript titled The use of social media and its impact on shopping behavior of Slovak and Italian consumers during COVID-19 pandemic to Sustainability journal. We really appreciate the time and effort that have dedicated to providing your valuable feedback on our manuscript. We are grateful to for your insightful comments on our paper. We have been able to incorporate changes to reflect most of the suggestions provided by you.

Round 2

Reviewer 2 Report

There seems to be significant improvement compared to the previous submitted manuscript. It is believed that the current manuscript can be accepted after a thorough proof reading of the paper.